# Occurrence of *Borrelia* sp. among Wild Living Invasive and Native Mesocarnivores in Poland

**DOI:** 10.3390/ani12202829

**Published:** 2022-10-18

**Authors:** Joanna Hildebrand, Kacper Jurczyk, Marcin Popiołek, Katarzyna Buńkowska-Gawlik, Agnieszka Perec-Matysiak

**Affiliations:** 1Department of Parasitology, Faculty of Biological Sciences, University of Wrocław, 51-148 Wrocław, Poland; 2Department of Tropical Medicine and Parasitology, Medical University of Gdańsk, M. Skłodowskiej-Curie 3a, 80-210 Gdańsk, Poland

**Keywords:** *Borrelia*, wild living carnivores, invasive species, zoonosis

## Abstract

**Simple Summary:**

Vertebrate hosts, especially wild living animals, are pivotal to the circulation and maintenance of *Borrelia* spp. Mesocarnivores are involved in Lyme disease ecology in sylvatic and suburban ecosystems. In this study, we aimed to examine the relative importance of six medium-sized carnivore species, raccoon, red fox, raccoon dog, European badger, pine marten and stone marten, as hosts of *Borrelia* spp. and investigated their role in this spirochaete’s transmission cycle. We also aimed to trace the reservoir competence of these invasive and native predators and the eco-epidemiology of *Borrelia* spp. in the context of a dilution effect. In all examined carnivore species, the occurrence of *Borrelia* was recorded, and the results suggest that raccoons may play a role as reservoir hosts for these spirochaetal bacteria. The role of invasive species seems to be worthy of further analysis with reference to the circulation of vector-borne pathogens as well as in the context of the “dilution effect” hypothesis.

**Abstract:**

Wild living mesocarnivores, both introduced and native species, are able to adapt well to peri-urban environments, facilitating cross-species pathogen transmission with domestic animals, and potentially humans. Individual tissue samples derived from 284 specimens of six carnivore species, i.e., raccoon, raccoon dog, red fox, European badger, pine marten and stone marten, were used for molecular investigations with the nested PCR method. The animals were sampled in the Ruszów Forest District (Poland). We aimed to examine the relative importance of the studied mesocarnivores as hosts of *Borrelia* spp. and investigated their role in this spirochaete’s transmission cycle. We also aimed to trace the reservoir competence of these invasive and native predators and borreliosis eco-epidemiology in the context of a dilution effect. The overall prevalence of *Borrelia* spp. in the tested carnivores was 8.8%. Almost all of the consensus sequences of the partial *flaB* gene shared identity with a sequence of specific *Borrelia* species, i.e., *B. afzelii*, *B. garinii* and *B. burgdorferi*. Our results suggest that raccoons may play a role as reservoir hosts for these spirochaetal bacteria. The role of invasive species seems to be worthy of further analysis with reference to the circulation of vector-borne pathogens as well as in the context of the “dilution effect” hypothesis.

## 1. Introduction

The genus *Borrelia* comprises arthropod-borne spirochetes which are pathogens of human and animals. Vertebrate hosts, especially wild living animals [1], are pivotal to the circulation and maintenance of *Borrelia* spp. The level of competence varies between hosts, with rodents appearing to be important hosts [2,3], whereas, e.g., roe deer (*Capreolus capreolus*) seem to be zooprophylactic or “dilution” hosts [4]. According to the dilution effect hypothesis, diverse host communities inhibit the parasite abundance through various mechanisms, such as regulating populations of susceptible hosts or interfering with parasite transmission [5,6,7]. Nevertheless, the role of carnivores as hosts in *Borrelia* spp. ecology remains unclear. The current knowledge of the complex interaction between this diverse pathogen and its vectors as well as different reservoir and non-reservoir hosts is still incomplete. 

Mesocarnivores such as raccoon (*Procyon lotor*), red fox (*Vulpes vulpes*), raccoon dog (*Nyctereutes procyonoides*), European badger (*Meles meles*), pine marten (*Martes martes*) and stone marten (*Martes foina*) serve as hosts for several tick species [8,9,10,11] and their significant involvement in Lyme disease ecology in sylvatic and suburban ecosystems is likely. The castor bean tick (*Ixodes ricinus*) is known as one of the main vectors of *Borrelia* spp. in Central Europe and has quite often been found on these animals [12,13,14].

In Poland, raccoons and raccoon dogs are considered invasive and alien species [15]. They may introduce new pathogens to the environment and may serve as potential hosts for many infectious agents important to domestic animals and humans. Some of the predators have a direct impact on the host competence, e.g., red foxes reduce, by hunting, populations of rodents, being one of the most numerous reservoirs of *Borrelia burgdorferi* s.l. [16].

Here, we aimed to examine the importance of six medium-sized native and invasive species of carnivore hosts of *Borrelia* spp. with overlapping ranges and to investigate their role in the spirochaete transmission cycle. We also aimed to check the reservoir competence of these predators and to study the borreliosis eco-epidemiology. 

## 2. Materials and Methods

### 2.1. Sample Collection

A total of 586 tissue samples (ear, spleen, liver) derived from 284 specimens of six carnivore species, i.e., raccoon (*n* = 51), raccoon dog (*n* = 50), red fox (*n* = 50), European badger (*n* = 53), pine marten (*n* = 27) and stone marten (*n* = 53), were used for molecular study (Table 1). The animals were sampled in the Ruszów Forest District (Poland), in the west part of the Lower Silesian Wilderness, being the largest lowland forest complex in Europe and the unique location of co-occurrence of the native and invasive carnivore species in Europe. The tissue samples obtained through collaboration with other projects were collected mainly during a predator control operation carried out as a part of the capercaillie (*Tetrao urogallus*) reintroduction program in the Lower Silesian Forest [17,18]. The tissue samples were delivered to the laboratory of the Department of Parasitology, University of Wrocław, and stored at −20 °C.

### 2.2. Molecular Analyses

DNA was extracted using the Bio-Trace DNA Purification Kit (EURx, Poland), following the manufacturer’s instructions. DNA concentrations were estimated with a NanoDrop 2000 spectrophotometer (Nanodrop Technologies, Wilmington, DE, USA). The molecular detection of *Borrelia* spp. was performed in three steps, i.e., (1) qPCRs on all obtained tissue samples of invasive (raccoon and raccoon dog) and native species (red fox); (2) nested PCRs targeting the *flaB* gene on selected samples of all carnivore species co-occurring in the study area; (3) conventional PCR targeting the 5S-23S rDNA intergenic spacer region on positive samples for the *flaB* gene.

Firstly, qPCR with the *ospA* gene as a marker and primers and probes such as B-OspA_modF, B-OspA_borAS and B-OspAmodPatto was applied [19,20]. The qPCR was performed on all tissue samples (ear, liver and spleen) obtained from raccoons, raccoon dogs and red foxes. The presence of *Borrelia* spp. DNA was determined by the amplification of a 600 bp fragment of the *flaB* gene in nested PCR, with two primer sets: 132f, 905r and 220f, 824r [21]. As the qPCR results were confirmed by nested PCRs, we performed only nested PCRs for other co-occurring carnivore species such as badgers and pine and stone martens (mustelids). In the last step, the *flaB-*positive samples were further examined using conventional PCR assays amplifying parts of the 5S-23S rDNA intergenic spacer region (IGS) with primers IGS_A and IGS_B [22]. Negative controls with nuclease-free distilled water as well as positive controls (*B. afzelii* from *Ixodes ricinus*) were included in each PCR reaction. In the PCRs, we followed the protocols described in the above-mentioned literature. 

All nested PCR-positive amplicons were purified using Exo-BAP (EURx) and sequenced in both directions by Macrogen (Amsterdam, the Netherlands) with the primers used for DNA amplification. The nucleotide sequences were edited using DNA Baser Sequence Assembly software (Heracle BioSoft, Mioveni, Romania) and compared to each other and with corresponding sequences deposited in GenBank using the Basic Local Alignment Search Tool (BLAST) program (http://blast.ncbi.nlm.nih.gov/Blast.cgi, accessed on 30 September 2022). The representative sequences were deposited in GenBank under accession numbers OP559180–OP559187.

## 3. Results

The samples obtained from skin (ear), spleen and liver of raccoons, raccoon dogs and foxes were checked for the presence of *Borrelia* DNA using qPCR and nested PCR (14 vs. 16). Apart from one case, all positive samples were obtained for ear isolates (16/151; 10.6%). The only other sample, which proved to be positive in the qPCR, originated from raccoon liver. Following the results obtained for raccoons, raccoon dogs and foxes, the nested PCRs was performed for co-occurring mustelids only. The overall prevalence of *Borrelia* spp. In all tested carnivores was 8.8%, considering skin samples (25/284) (Table 2), and the infection rates varied significantly between these host species. Almost all consensus sequences of the partial *flaB* gene which were analyzed with the BLAST method revealed shared identity with a sequence of specific *Borrelia* species, i.e., *B. afzelii* (23), *B. garinii* (1) and *B. burgdorferi* (1) (Figure 1).

The highest prevalence (23.5%) was observed for raccoons (12/51) as compared to raccoon dogs and red foxes with a single positive sample (prevalence of 2% each) (Table 2). The majority of positive samples were *Borrelia afzelii* (100% identity with, e.g., CP018262, DQ016619, KF894068, KX646195), one sample from fox was identified as *B. garinii* (100% homology with a few sequences, i.e., KF894068, MF150061).

The badgers proved to be more susceptible to *Borrelia* infection. Of 53 tested skin samples, eight were positive (15.1%), while DNA of *Borrelia* sp. was found in two samples obtained from pine marten (27 tested, i.e., 7.4%) and one from stone marten (53 tested, i.e., 1.9%). The BLAST analysis of seven sequences showed 100% similarity to several *Borrelia afzelii* sequences; one was 100% identical to a *Borrelia burgdorferi* sequence (acc. no. KF836508) from *Ixodes ricinus* from Poland (Lower Silesia) and 99.8% identical to other sequences (KX646200, KR782218) from *I. ricinus* from Poland as well as one sequence (acc. no. CP077727) from Belarus.

The interspecific divergence of IGS sequences of *Borrelia afzelii* was 0.0–2.3% (0–6 bp out of 264 bp), and three groups could be distinguished when analyzing the nucleotide similarity. The comparison between these lines and the homological sequences available in GenBank showed 100% to 99.2% similarity to samples derived from humans (CP018262, CP002933, JX888444), *Ixodes* ticks from Russia, Estonia, Germany (CP009212, KX418638, AY772053, MW489224, OL848283) as well as to sequences obtained from rodent tissues from France and Romania (KY273112, KY123663).

## 4. Discussion

The majority of data on *Borrelia* spp. in raccoons originate from the USA and are based on the results of serological testing [23,24,25,26]. Additionally, the results of studies carried out by Tufts et al. [27] showed the presence of *B. burgdorferi* s.l. in 2.6% of examined samples. Serological studies concerning this spirochaetal infection in raccoons from introduced areas, carried out in Japan, revealed the presence of *B. afzelii* (0.1%) and *B. garinii* (0.1%) [27,28]. Our results, with the first molecular evidence of *B. afzelii* infection, attaining 24.0% in the European population of introduced raccoons, indicate this carnivore as a potential reservoir host for the aforementioned spirochaetal species. 

Molecular studies carried out by Wodecka et al. [10] on European raccoon dogs in Poland revealed that 11.9% of tested animals were positive for *B. garinii* (dominant species), followed by *B. afzelii* and *B. valaisiana*. Our survey indicated a much lower level of *Borrelia* infection—2.0% in raccoon dogs, being detected as *B. afzelii* only. On the other hand, a study carried out in South Korea, on native carnivores, resulted in the first report of *Borrelia theileri* (0.7%) in raccoon dogs [29].

In the present study, badgers yielded a higher prevalence of *Borrelia* spp. (15.1%) compared to other tested species of native carnivores, although *Borrelia afzelii* and *B. burgdorferi* were detected as well. Research by Wodecka et al. [10] provided evidence for *Borrelia* infection in badgers with a similar level of prevalence. Analyses of all PCR-positive blood, ear biopsy and liver samples revealed that 12% of badgers were infected with borreliae. Gern and Sell [30] detected the presence of both *B. afzelii* and *B. valaisiana* in 19.4% of tissue samples from the ears of badgers in Switzerland. The PCR-positive tissue samples identified in badgers revealed that they were infected exclusively with *B. afzelii*. In contrast, during research conducted in Belgium and the Netherlands, *Borrelia burgdorferi* (s.l.) was detected in 0.9% of liver samples of European badgers [11]. 

DNA of *B. burgdorferi* (s.l.) was found in tissues of other musteloid species, i.e., stone and pine martens. *Borrelia burgdorferi* (s.l.) was detected in 3.9% of spleen samples of pine martens and 2.9% of ear biopsies of stone martens [11]. To the best of our knowledge, this study is the first molecular identification of *B. afzelii* in stone marten (7.4%) and in pine marten (1.9%).

To date, there have been only a few reports from Europe on the occurrence of *B. burgdorferi* s.l. in tissues of red foxes in Germany [31,32], Romania [33], Norway [34] and Poland [35]. Results of our study showed that the prevalence of *Borrelia* spp. was 2.0% and *B. garinii* was the only species detected in red foxes in this survey. *Borrelia garinii* was also solely noticed in foxes in Germany, but the recorded prevalence was much higher—24% [31]. In Poland, the analysis of tissues of 243 animals showed that 23.5% of them contained DNA of *Borrelia* spp., whereas *B. garinii* was identified in 91% of the infected foxes [35]. On the other hand, only *B. afzelii* and *B. burgdorferi* were detected in tissues of red foxes in Romania [33].

## 5. Conclusions

To the best of our knowledge, this is one of the few and the most comprehensive studies undertaken to assess the importance of introduced and native carnivores in the ecology of *Borrelia* spp. worldwide. In all examined carnivore species, the occurrence of *Borrelia* was recorded. We identified *B. afzelii* (the most abundant genospecies), *B. garinii* and *B. burgdorferi* in studied animals. The highest level of *Borrelia* prevalence estimated in raccoons suggests they can play a role as reservoir hosts for these spirochaetal bacteria. The significance of invasive species is worth further analysis, with special reference to the circulation of vector-borne pathogens as well as in the context of the “dilution effect” hypothesis.

## Figures and Tables

**Figure 1 animals-12-02829-f001:**
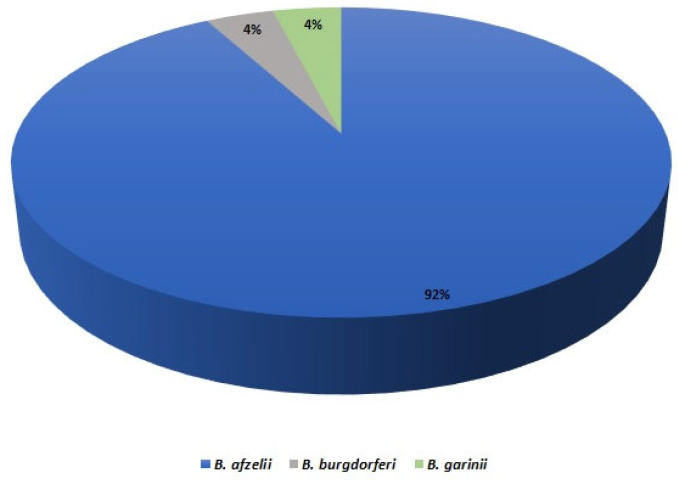
Distribution of *Borrelia* species identified in examined carnivores.

**Table 1 animals-12-02829-t001:** Tissue samples of carnivore hosts used in the present study.

Carnivore Host	Total No. of Tested Samples	No. of Sample Types		
		Ear Biopsy	Spleen	Liver
*Procyon lotor*	153	51	51	51
*Nyctereutes procyonoides*	150	50	50	50
*Vulpes vulpes*	150	50	50	50
*Meles meles*	53	53	-	-
*Martes martes*	27	27	-	-
*Martes foina*	53	53	-	-
Total	586	284	151	151

**Table 2 animals-12-02829-t002:** *Borrelia* species identified in samples obtained from wild living mesocarnivores based on *flaB* gene.

Carnivore Host	No. of Tested Specimens	No. of Positive Ear Biopsy Samples	Prevalence (%) (95% Confidence Interval)	Detected *Borrelia* Species (No. of Sequenced Samples)
*Procyon lotor*	51	12	23.5 (15.9–33.1)	Ba (12)
*Nyctereutes procyonoides*	50	1	2.0 (0.1–10.7)	Ba (1)
*Vulpes vulpes*	50	1	2.0 (0.1–10.7)	Bg (1)
*Meles meles*	53	8	15.1 (8.8–23.8)	Ba (7), Bb (1)
*Martes martes*	27	2	7.4 (1.3–23.7)	Ba (2)
*Martes foina*	53	1	1.9 (0.2–7.4)	Ba (1)
Total	284	25	8.8 (6.3–12.2)	Ba (23), Bb (1), Bg (1)

Ba—B. afzelii; Bb—B. burgdorferi; Bg—B. garinii.

## Data Availability

The data presented in this study are available on request from the corresponding author.

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
