# Peer review of "Occurrence of Borrelia sp. among Wild Living Invasive and Native Mesocarnivores in Poland"

_animals, 2022, doi:10.3390/ani12202829_

Round 1

Reviewer 1 Report

The results should be larger and more conclusive in Q1 journal. There is only one figure, they should try to provide more information about it.

Figure 1. Put the scientific names of the animals. The tables and figures must be self-explanatory. How many animals have they used? Is it a mean? Ls-means? There is no standard error? No significant differences? What does this figure contribute more with respect to the text?

In the objective of the work they assure the following: “we aimed to examine the relative importance of six carnivora medium sized species as hosts of Borrelia spp. and investigated their role in this spirochaete transmission cycle. We also aimed to trace the reservoir competence of these predators and borreliosis eco-epidemiology in the context of a dilution effect.” However, only 5 different species appear in the graph.

Introduction. Why are those five carnivores used? They should try to better explain that decision so that it does not seem arbitrary.

Material and methods. The sum of the samples (51+50+53+27+53) is 234, which does not agree with the 284 they claim.

There is no differentiation of how many of each of these samples belong to each of the different tissues. It is not specified if they have taken more samples from the same animal and what criteria have been followed.

Check the articles, there are some sentences in the results section “The isolates obtained from skin (ear), spleen and liver were analyzed for the presence of Borrelia DNA using the flaB gen as marker…,” These are not the results of the work. Same comments in point 3.2.

What are the concrete results of the work?

Reviewer 2 Report

The manuscript entitled ‘Wild living mesocarnivores, Borrelia sp., "dilution effect" - molecular studies’ by Hildebrand J. et al. investigates the role of six carnivora medium sized species in the circulation of Borrelia spp. However, manuscript (MS) present important results, The MS needs substantial editing. The materials and methods, the results part respectively the discussion must be revised, even the conclusions. Until these fundamental issues are not revised, I cannot recommend the acceptance of the MS.  I propose to adapt this research article to a short report/communication.

Simple summary/Abstract

 L17: revise the sentence. The eco-epidemiology of Borrelia spp. and not of disease.

L18/L32: may play a role

L19/201: revise the sentence. Delete ‘in our opinion’

 Introduction

L42: delete ‘most important’, revise the sentence.

L43-46: revise the sentence, the message of this sentence in cannot be understood.

L47-49: this sentence must be moved first before sentence L47.

L62: in Borrelia spp. ecology and not Lyme disease

Materials and methods

L79-82: enumerate the species name of animals

It is not clear why different genes were targeted. Why not only qPCR was used for molecular screening for all of the samples? Explain this, make it clear

Results

L124, 134: the infection prevalence in %

It will be much easier to follow the results in a table

L119. 144:  do you mean B. burgdorferi s.s.?

Discussion

L163-164, L170: put 0.1% in () or revise the sentence.

L172: (29)?

L174-175: where in the results B. burgdorferi s.s. is mentioned? Revise the sentence in the results section

L191: “B. garinii was the only species detected in this survey”????

Reviewer 3 Report

General comments:

Manuscript entitled "Wild living mesocarnivores, Borrelia sp., "dilution effect" - molecular studies" describes the occurrence of Lyme disease (LD) Borrelia species in six mesocarnivore species (racoon, racoon dog, red fox, European badger, pine marten, and stone marten). The dilution effect mentioned in the title is indicated only in the Introduction, but not in the Results and Discussion. In my opinion, it should be replaced by the occurence of invasive species that were examined in the study and are worthy to discuss in detail. The manuscript could be interesting without attention-getting but confusing phrases like mentioned "dilution effect". 

Below there are minor comments that should be taken into account before publication of the manuscript.

Minor comments:

The whole text:

1) the name "Borrelia" should be in italics.

Simple summary:

line 13 - there is "carnivora", it should be "carnivore" or "Carnivora" (order's name).

Abstract:

line 26 - change the phrase "used to for molecular investigations using the nested PCR method" to "used for molecular investigations with the nested PCR method".

line 27 - "studied mesocarnivores" instead of "examined mesocarnivores".

Introduction:

lines 60-61 - the proper latin names of (European) pine marten and stone (beech) marten is Martes martes (not Martens martens) and Martes foina (not Marten foina), respectively!!!

lines 61-62 - replace the phrase "there exists the probability" by the simple "there is probability".

line 67 - may racoons and racoon dogs be the vectors of any pathogens?

Materials and methods:

lines 86-87 - "Tetrao urogallus" should be in italics. 

lines 99-100 - what was the positive control in the study?

line 107 - the sequences may be "deposited" but not "deposed" in GenBank.

line 110 - the accession numbers should be given.

line 111 - "The study was performed in three steps...?" And four steps are below-mentioned. So, what is the truth?

Results:

line 128 - "...100% homology..." - this is incorrect term, homology has no numerical value, the term "identity" should be used.

lines 133-135 - in the sentence "Apart from one case, all positive samples....(16/284),..." - the data concern the Procyonidae and Canidae family members (racoons, racoon dogs and foxes) that add include only 151 specimens not 284. Please, correct the data.

line 153 - the name "Ixodes" should be in italics.

Round 2

Reviewer 1 Report

The authors have carried out all my suggestions. Explain and define graphs and tables so that they are self-explanatory.

Author Response

Dear Reviewer,

Thank You. According to your suggestion, the manuscript has been modified and clarified in some fragments (graph and table) to be better understood. In addition, the manuscript has been corrected and edited by a colleague fluent in English writing.

Reviewer 2 Report

I have no more comments and suggestions to the authors, except the English language revision. 

Author Response

Dear Reviewer,

Thank You. According to your suggestion, the manuscript has been corrected and edited by a colleague fluent in English.